# Wapekeka’s COVID-19 Response: A Local Response to a Global Pandemic

**DOI:** 10.3390/ijerph191811562

**Published:** 2022-09-14

**Authors:** Keira A. Loukes, Stan Anderson, Jonas Beardy, Mayhève Clara Rondeau, Michael A. Robidoux

**Affiliations:** 1School of Outdoor Recreation, Parks, and Tourism, Lakehead University, Thunder Bay, ON P7B5E1, Canada; 2Indigenous Health Research Group (IHRG), University of Ottawa, Ottawa, ON K1N6N5, Canada; 3Wapekeka First Nation, Angling Lake, ON P0V1B0, Canada; 4Faculty of Law, University of Victoria, Victoria, BC V8P5C2, Canada; 5School of Human Kinetics, University of Ottawa, Ottawa, ON K1N6N5, Canada

**Keywords:** First Nations, Indigenous health, COVID-19 response, food security, food sovereignty, First Nations governance, subarctic Ontario, traditional food

## Abstract

Two years after the onset of the COVID-19 pandemic, many nations and communities continue to grapple with waves of infection and social fallout from pandemic fatigue and frustration. While we are still years away from realizing the full impacts of COVID-19, reflecting on our collective responses has offered some insights into the impact that various public health policies and decisions had on nations’ abilities to weather the multifaceted impacts of the pandemic. Widely believed to have the potential to be devastated by COVID-19, many Indigenous communities in Canada were extremely successful in managing outbreaks. This paper outlines one such example, Wapekeka First Nation, and the community’s formidable response to the pandemic with a specific focus on food mobilization efforts. Built on over a decade of community-based participatory action research and informed by six interviews with key pandemic leaders in the community, this paper, co-led by two community hunters and band council members, emphasizes the various decisions and initiatives that led to Wapekeka’s successful pandemic response. Proactive leadership, along with strong traditional harvesting and processing efforts, helped to take care of the community while they remained strictly isolated from virus exposure.

## 1. Introduction

The COVID-19 pandemic and its devastating health and economic impacts continue to be felt unevenly around the world as varying health, economic, political, and geographic conditions have made certain groups of people more vulnerable to the virus than others. Indigenous communities in Canada, especially those in rural and remote regions of the country, were seen as particularly vulnerable to COVID-19 infection for many reasons, most notably limited health resources and services, limited infrastructure (such as housing), and a disproportionately high prevalence of comorbidities [1,2,3,4]. The term “Indigenous” in Canada refers to First Nations, Métis and Inuit peoples, all living within intricate cultural systems connecting each Nation to its territory and homeland [1]. Each community within each Nation is distinct and unique (i.e., there are over 630 First Nations communities who speak over 50 different languages. The increased vulnerability to COVID-19 in Indigenous communities has been directly caused by historical and contemporary colonial policies which have disrupted traditional Indigenous food, governance, language, and knowledge transmission systems [5]. While early on in the pandemic, Indigenous communities did not experience high infection rates as was expected [6], as of 8 March 2022, COVID-19 infection rates in Indigenous communities were 2.1 times the respective rate of the Canadian population [7]. The fatality rate, however, was 73% that of the rest of the country, with 96% of those infected having recovered [7]. This could be attributed to timely, effective, and prepared community responses to COVID-19 in many Indigenous communities [6,8], such as organizing food packages, harvesting traditional medicines [9], mobilizing funding from external networks [10], and by controlling community borders [11]. Despite these successes, stories of COVID-19 responses in Indigenous communities are not regularly acknowledged or shared [12].

In this article, our objectives are to highlight the formidable response of the Wapekeka First Nation, an Oji-Cree community in northwestern Ontario, to the potentially devastating impacts of COVID-19. Written alongside two hunters from the community (S.A. is Deputy Chief and J.B. is an elected Band Member) this article highlights food mobilization efforts in the community and their impact on food security and food sovereignty in the region (K.A.L. comes from German, Scottish, and Anishinaabe relatives; M.C.R. is a member of Moose Cree First Nation and of French-Canadian background, and M.A.R. is a settler scholar and ally to Indigenous land rights and traditions. All IHRG members are visitors in Wapekeka First Nation).

### 1.1. Colonial Impacts on Indigenous Health

Contact with Europeans in northern Ontario started in the early 1600s with the development of trading posts on the Hudson’s Bay coast. Prior to contact, Cree, Anishinaabe, and Oji-Cree families in the region tended to travel in small kinship groups of 10–15, following large mammals as a food source [13,14,15]. The presence of the trading posts attracted more families to gather around their vicinity, trading furs and traditional food for European goods and food. The growing populations around these forts put more pressure on food resources in the surrounding areas. These pressures were further intensified with the development of the Canadian Pacific Railway, enabling non-Indigenous hunters to have access to the north to hunt game, most specifically, beaver [16]. The introduction of capitalism through the commercialization of the hunt along with increased populations of Indigenous and non-Indigenous peoples around the forts led to an eventual decline in the population of large land mammals. Without adequate access to food, many community leaders in the region decided to enter into Treaty with the Crown in order to receive protection, healthcare, and education from the Canadian government (Long, 2010). Treaty 9 was signed in 1905–1906, although Wapekeka First Nation did not enter Treaty 9 until signing the adhesion of 1929–1930. The Treaty signing process, however, did not involve any negotiations—it was written and signed by the Provincial and Federal governments before arriving in the community, and evidence suggests Indigenous leaders were intentionally misled by commissioners [17,18,19,20].

Before the signing of Treaty 9, the *Indian Act* (1876) was already in full effect in Canada. This race-based policy continues to govern Indigenous people today, ultimately disrupting Indigenous nationhood [21]. The *Indian Act* was designed to assimilate Indigenous peoples into European society and can be characterized as an act of cultural genocide [22,23,24]. Among other control measures, the *Indian Act* limited mobility of Indigenous peoples, dismantled traditional governance systems by replacing them with colonial-style Band Councils, regulated who has “Indian status” and who does not, and mandated all school-aged children attend Indian Day Schools (IDS) and Indian Residential Schools (IRS) managed by Catholic, United, Presbyterian, and Anglican churches. Over 150,000 “Treaty-Status” youth between the ages 7 and 15 attended these schools [25]. The IRS system subjected generations of Indigenous youth to physical, sexual, and psychological abuse [24] as well as medical and nutrition experiments [5,26] resulting in devastating physical, emotional, spiritual, and mental trauma [27]. The intergenerational impacts of the residential school system are affirmed by survivor statements documented within the Truth and Reconciliation Commission (TRC) and by scholars across settler-colonies [28,29,30,31]. For example, it has been demonstrated that prior stressors and early life trauma can contribute to an increased vulnerability to pathology [32]. As such, many of these traumas reverberate to this day, exemplified through the gap in socio-economic health disparities between Indigenous and non-Indigenous peoples [32].

Internationally, the impacts of colonization have been recognized as a key determinant of health for Indigenous peoples [33]. In Canada, multiple studies demonstrate how Indigenous communities experience a higher burden of chronic disease and higher rates of mortality related to service provision, poverty [8], chronic underfunding and weak public infrastructure [34]. For example, despite being more vulnerable to COVID-19 due to comorbidities, Indigenous peoples (who make up 4.9% of the population) received only 1% of federal funds at the start of the pandemic [35]. Some First Nations have stated that funding has declined since June 2020, drastically limiting mental health supports [36]. Public health guidelines for COVID-19, which focused on hand washing, social distancing and COVID-19 testing as preventative measures, required access to clean running water, the ability to socially isolate, and adequate housing and healthcare infrastructure—resources which are not readily available in many First Nations in northern Ontario (and throughout much of Canada) [3,5]. In this way, the early years of the pandemic emphasized long-standing inequalities. For example, within Nishnawbe Aski Nation (NAN), a political entity that represents 49 First Nation communities in northern Ontario, seven communities are under short term boil water advisories, and thirteen communities are under long term boil water advisories [37] (a “boil water advisory” is put in place by water treatment plants when the water has or could have bacteria or other pathogens that would be considered unsafe to drink). Thirty of the forty-nine communities face considerable infrastructure and access challenges as they are remote fly-in communities without permanent road access. Health services in these communities are provided through nursing clinics with rotating doctors and nurses [38]. Emergency and complex medical procedures require patients to be flown out to Sioux Lookout, Thunder Bay, or Winnipeg. Despite Indigenous healthcare being a Treaty obligation and inherent right [39], it is still insufficiently funded and supported by the Canadian government. This has led many communities to approach health crises on their own.

### 1.2. Indigenous Governance of Indigenous Health

Historically, Indigenous peoples have endured waves of infectious disease brought by European settlers to which they had no natural immunities, such as smallpox, tuberculosis, influenza, and measles throughout the 19th and 20th centuries [40]. Throughout these crises, community intergenerational knowledge has been preserved through the creation of holistic and diverse pandemic responses, which are central to the ongoing “survivance” of Indigenous communities [8]. During the COVID-19 pandemic, many First Nations communities exercised their inherent rights to self-government by creating community-specific protocols to control infection rates [41]. For example, Cree Artist Donna “Dolores” Gull drew on her cultural strengths and teachings by designing and sewing over 60 ceremonial masks stuffed with cedar and sage for her community [42]. Due to rising food prices, many Indigenous communities shared traditional foods through community-based food markets [36]. For example, Moose Cree First Nation endeavored to provide traditional food to their members by recruiting volunteers to assist in moose hunting and fishing [43]. In Whitesand First Nation, community members delivered food hampers, cleaning supplies and craft supplies to those required to self-isolate [42]. These community-driven, nation-specific responses to the COVID-19 pandemic demonstrate immense community collaboration. Wapekeka First Nation’s response is no exception to this.

### 1.3. Community Profile: Wapakeka First Nation

Wapekeka First Nation is a subarctic Oji-Cree community in northwestern Ontario, located 451 km northeast of Sioux Lookout (see Figure 1), with a population of 440 [44]. Geographically remote, Wapekeka is accessible year-round by plane and by ice roads after freeze up, making the community heavily reliant on air-freight for access to goods such as market food. Wapekeka is connected by a year-round dirt road to Kitchenuhmaykoosib Inninuwug (KI) (Big Trout Lake), a community of 1025. In ideal conditions, it is a 60-min round trip to KI, which is important to Wapekeka, as it provides access to grocery stores with greater variety and selection. Wapekeka’s community infrastructure includes residential housing, a small independent grocery store, two confectionary stores, a gym, a band council hall, a nursing clinic (with rotating nurses) and a primary school. Currently, the community is building a new school that will include a high school (to Grade 10), allowing the youth to stay in community beyond primary school instead of being required to stay with billet families to attend high school in Thunder Bay or Sioux Lookout.

## 2. Research Methodology and Methods

This paper builds on the Indigenous Health Research Group’s (IHRG) 16 years of strong relationships and local efforts in northern Ontario First Nations including Wapekeka First Nation. The IHRG is a multidisciplinary team of researchers working with First Nations leadership to address community needs, concerns, and solutions to ongoing food security challenges. The IHRG, along with four partnering First Nations communities, are supported by a five-year Social Sciences and Humanities Research Council of Canada (SSHRC) Insight Grant to explore and understand local food security challenges and to support local food initiatives. To foster a research process that is collaborative and holds Indigenous perspectives and ways of knowing at its core, this project is guided by Indigenous methodologies (IM) and community-based participatory research (CBPR). Philosophically, IHRG works from the perspective that “the people know and can reflect on their own life, have questions and priorities of their own, [and] have skills and sensitivities which can enhance (or undermine) any community-based projects” (Smith 1999, p. 127). Participatory action research such as CBPR and IM necessitate that the research design, questions, analysis, and dissemination are conducted *with* and *by* the community. Research conducted with these methodologies at its core works from the position that communities themselves, not outsider experts, are best situated to develop solutions to the challenges their members identify [45,46,47,48,49]. IM emphasizes authentic, long-term relationships building that centers Indigenous community members and ways of knowing in the research process [50,51]. This process also helps ensure that the community’s interests are prioritized and that access to sensitive material is appropriately guarded as highlighted by the First Nations Principles of OCAP^®^ (Ownership, Control, Access and Possession) [52]. It is important to note that this research is conducted according to the University of Ottawa’s Research Ethics Board Guidelines (H-05-18-607), The First Nations Principles of OCAP^®^, and the community’s specific research protocols.

Due to COVID-19 restrictions, our research team was unable to travel to partnering communities in 2020 and for most of 2021. During this time, the last author (M.A.R.), who maintains relationships with community members through text and Facebook messenger, noticed the pictures shared of the incredible traditional food harvesting, processing, and distribution that took place in Wapekeka. The community has long been strategizing how to increase traditional food harvesting and distribution while encouraging more young people to become interested in attaining these skills. By focusing on food mobilization amid a global pandemic, the community achieved precisely this.

To get the full picture, the second (SA) and third author (J.B.) invited the first (K.A.L.) and last (M.A.R.) author to visit Wapekeka to meet with a few key leaders of the COVID-19 response team during a week-long window between the hunters’ return from their fall camps and governance meetings in Thunder Bay. This coincided with a brief window in October 2021 when COVID restrictions were eased due to the increased rate of vaccinations and low case rates. After 18 months of travel bans, we were eager to take the opportunity to reconnect with friends and colleagues. K.A.L. and M.A.R. met with S.A. and J.B. in Thunder Bay and Wapekeka to strategize who to speak with throughout the week. During those seven days in the community, in between dinners with friends, volleyball games, partridge hunting, and setting beaver traps, five community members shared their experiences in semi-structured interviews (we spoke to a sixth participant over the phone upon our return to Ottawa). This time spent visiting outside of semi-structured interviews is an extremely important part of community-based participatory research and Indigenous methodologies [53], and led to many informal conversations that helped create a full story. Intentionally, two interviewees were men and four were women, as we wanted to place a greater emphasis on Indigenous women’s perspectives which are often missing from research on food mobilization [54,55,56]. Since J.B. was a key leader in the food mobilizing efforts during the COVID-19 pandemic, we decided to “interview” him as well, and include some key quotes, as his front-line experiences in the field were invaluable contributions to understanding the experiences of the harvesters.

All participants in this study provided informed consent to participate in this paper. Both K.A.L. and M.A.R. were present at each interview. The semi-structured interviews were audio recorded with those participants who felt comfortable with it, which was dependent on our previous relationships. Where the interviews were not recorded, one interviewer would prompt the conversation and the other would take notes. All interviewees were given the option for anonymity, and all but the Wapekeka Pandemic Committee members chose to use their real names. We called those who chose to remain anonymous “a Wapekeka Pandemic Committee member” in order to contextualize the quotes while maintaining anonymity. The audio recordings were transcribed and read over multiple times to achieve immersion in the data. The interviews corroborated each other and together told a more complete story of the various layers of Wapekeka’s pandemic response. We intentionally chose to use longer direct quotes to ensure the voice of each participant came through, as this is a story of Wapekeka’s response according to the community members, not outside perspectives. After piecing the narrative together, the first draft of the paper was reviewed, assessed, and commented on by all authors who approved and suggested revisions. What follows is a story of the pandemic response, pieced together through six different stories and perspectives of Wapekeka’s community-driven, proactive, and innovative pandemic response.

## 3. Results and Discussion

### 3.1. Proactive Community Leadership


*“If you live in the community, you know what works and what doesn’t. I don’t try to bring someone else’s plans into my community, and if I do, we integrate them with what we know is best.”*


-
*Wapekeka Pandemic Committee member, Wapekeka*


In March of 2020, and in the months preceding, people all over the world were planning their response to a virus about which little was known. There were many unanswered questions about the spectrum of severity, mechanism of spread, rate of infection, potential for vaccine development, capacities of health care systems, resilience of global supply chains, and frequency of community-wide lockdowns. While these fears and uncertainties were shared globally, a Wapekeka Pandemic Committee member knew that local COVID-19 outbreaks or extended lockdown measures would have dire consequences in her remote, northern First Nation community:

For sure there was a lot of talk within the community. When I first heard of it personally, I mean, where it all escalated from—and after the previous pandemic of H1N1, when we lost one newborn baby—I know that I tried, or we tried our best to prepare for the next one because we have a plan. So we started listening in, you know, just watching how far it’s escalating to our area, right? So we had meetings. I know everyone panicked.

Wapekeka does not have permanent health care staff (doctors and nurses fly in on a rotational schedule) nor does it have a fully stocked grocery store—it depends on all year access to KI’s stores. Although Wapekeka is relatively isolated, this Wapekeka Pandemic Committee member, along with the rest of the community, did recognize that with the vast number of people coming in (rotating nurses, construction workers, and other professionals) and out (medical appointments) of their community, it was only a matter of time before the virus made its way there. Additionally, any impact in global food supply chains is felt exponentially in remote northern Ontario First Nations. In addition to the challenges of shipping market foods from the south to northern remote regions, Indigenous peoples’ historical experiences with colonial governments has taught many First Nations that they will not be top priority in national crises. For example, during the H1N1 pandemic when First Nations asked for help to manage the outbreaks they were experiencing, the Canadian government infamously sent body bags and a package of masks and hand sanitizers to two remote First Nations in Manitoba [57,58]. Chief Jerry Knott (Wasagamack First Nation) asked at the time “Is the body bags a statement from Canada that we as First Nations are on our own?” [58].

Wapekeka’s leadership knew that they would need to initiate their own response to take care of their community instead of waiting on federal or provincial actions. Taking Provincial and Federal measures into consideration, the Wapekeka Pandemic Committee largely planned their own pandemic action plan. Their records and plans from earlier crises guided their initial response, although the committee knew they needed to be extremely flexible in order to keep up with the ever-evolving science around the COVID-19 virus. As a Wapekeka Pandemic Committee member stated, “when we started working with our plan, there’s always changes to the plan, every week… there’s always going to be a change to it, because the virus changes”.

In March 2020, the Wapekeka Pandemic Committee held six to eight meetings a day with the band council, nurses, community, Nishnawbe Aski Nation (NAN), and Sioux Lookout First Nations Authority (SLFNA). Although these broader governing bodies shared information and guided protocol development, Wapekeka’s response was self-led, independent of external financial support, and consciously employed protocols that were stricter than those advised. A Wapekeka Pandemic Committee member stated:
I’ve always said from the beginning, each community is different. They work-differently. If you live in the community you know what works and what doesn’t, and that’s what you have to work with. I don’t try to bring someone else’s plans into my community, and if I do, we integrate them with what we know is best.
Once plans were put into place, the pandemic Wapekeka Pandemic Committee focused on equipping the community with adequate PPE and sanitation supplies. The same person explained:
So the community set up a station at the gym where they did all the food preparation. The flights were closed. Nobody was coming in or going out unless it was an absolute medical emergency, meaning a med-evac. I know there was a lot of fear, there was a lot of stigma. Everyone worked hard. A lot of people or, most people, had to stay home and work out of homes. And even though with the limited housing, we have multiple families in one household, the focus at that time was to get to keep the virus out. So there were times everything was—we couldn’t get anything, we couldn’t get any masks: they were all sold out. You know, all the cleaning supplies, PPE, stuff like that. We had to fly in all the stuff that we needed, chartering freight and chartering PPE or whatever we could find out there…The community membership didn’t want nurses or any other professionals that came to the community to travel through major airports, like Toronto and Montreal, you know, all the way going west.
A priority for the Wapekeka Pandemic Committee was ensuring COVID-19 stayed out of the community. This meant banning all travel in and out of the community unless for essential (i.e., medical) purposes. If people did have to leave, they were required to isolate for 14 days upon their return with everyone in their household in one of four houses on the outskirts of the town. A team of women volunteers were responsible for cleaning and setting the homes up with beds, bathroom supplies, washers and driers, and refrigerators, while other community members were responsible for running errands for the family isolating.

All of these measures were undertaken with the Band’s budget, which has just started being compensated from the government. The same Wapekeka Pandemic Committee member expressed mixed feelings about the government’s response in the pandemic:
Actually, in ways I was happy and some ways disappointed. For example, as soon as the vaccine was available, the communities’ memberships were prioritized to get it first. But in other ways it was just talk and no action. It’s just how it is with government…I think most of it, we’re getting the money back: it’s just a process.
When we asked the Wapekeka Pandemic Committee member whether federal or provincial guidelines made any difference in how Wapekeka made decisions regarding protocols and guidelines during the pandemic, the person paused and then replied:
No, there was no immediate action that—there might have been words that were said and that was it. But, I know they tried their best as soon as we started to figure out what was running low and what we needed because we couldn’t find it. We got outside resources that helped and we got our PPE supplies flown in from other organizations such as NAN and SLFNA.

Community meetings and Facebook posts were an important means of communication to update membership on pandemic protocols and actions. Facebook is a powerful and popular communication tool throughout the north. In Wapekeka, Facebook pages and groups such as the “Wapekeka Pandemic Plan and Events” and the “Wapekeka Pantry/Harvestry” were used to communicate pertinent information regarding the pandemic. This platform was essential for Wapekeka’s mock drills, where a designated Wapekeka Pandemic Committee member would post that there was a confirmed positive case in the community. Everyone would be updated immediately and carry out community lockdown measures. These emergency lockdown procedures were followed by everyone in the community. These mock drills were extremely effective in helping the community prepare—one family recounted their six-year old’s response upon being the first in the household to hear of a positive case in KI. He and his friends immediately followed the protocols they had practiced in the mock drills—they put on their masks, went home, and told their families to prepare for a lockdown. The family laughed as they recounted the story, yet their pride in the preparedness of their children was clear.

The first lockdown in March 2020 caused tremendous stress to the community as all flights were stopped and the road to KI was often closed. Following this, any time there was a reported positive case in Wapekeka or KI, there was a lockdown. The Wapekeka Pandemic Committee member explained the intimate relationship Wapekeka has with KI:
We had to work with KI closely, our neighbouring community, because we have an all-season road [connecting the two communities]. The plane stops in KI and here, and if KI gets a positive [case], we have to notify whoever was on that flight. That’s how it worked, that’s how we communicated and worked together. But in the meantime, we had to close the road; there was no traffic coming or going except the police.
Overall, Wapekeka had a total of four cases, all of which were from the same household. When this family was in quarantine, the community would bring them food, supplies, and “home brewed” traditional medicines. A Wapekeka Pandemic Committee member explains:
Yeah, there were—even before the pandemic they were already practicing traditional medicines through our land-based programming. So they were already teaching that, people were learning, young ages, even with me. I knew what to get, because it was taught; I never knew it before. And that’s what most people use even today. They would talk about it with the older people, were talking about when they first were hit with the—they said they had a pandemic before, I don’t know if it was the Spanish, or whatever, from what I could gather is that everyone got sick, half of them died, but there was only one person that never got the virus, or a handful of people, and those were the ones that took care of the people within the community. They said this was expected they were saying and there’s more to come.
This example of the community coming together to care for one another, a trend that quickly became a signature of Wapekeka’s COVID-19 response, stands in contrast to the experiences of many families and individuals in much of the rest of Canada which tended towards more insular and individual approaches. For example, the Angus Reid Institute recently published an online survey of 2550 Canadians which revealed that 82% of respondents believe the pandemic has pulled people further apart, 79% believe it has brought out the worst in people, and 61% believe their level of compassion for one another has grown weaker [59,60,61].

Wapekeka’s lockdowns changed household food systems immensely as many families either rely on flights to Thunder Bay or Sioux Lookout to purchase food and other supplies for the year or drive to the larger grocery stores in KI to have access to greater variety and supply of groceries. Although shutting down travel was essential to keeping the virus out, it also made food supply for many families in the community more precarious. Initial pandemic panic created a purchase rush in the community as people tried to secure food for their families. This led the Wapekeka Pandemic Committee to come up with a plan in order to ensure the community continued to have access to both market food and traditional food. A Wapekeka Pandemic Committee member described how the plan came together:
I knew the store couldn’t handle, wouldn’t be able to provide all the necessities for the community, that’s why we had opened up the gym and got our supplies in. And every household got a hamper. And it’s not just food, it’s diapers, baby formula, hygiene care, cleaning supplies, and PPE… And it’s not just only that, we had to buy freezers, fridges to pack up all the meat and put it into those containers. All the stuff like that. And we only had so many people on the team because we were trying to limit the number of people in one room area. What we lacked was space, a lot of space.
The Wapekeka Pandemic Committee were not the only ones making plans to provide food for the community.

### 3.2. Mobilizing Traditional Food Systems

#### 3.2.1. Harvesting


*“You gotta be cautious when you’re out there, it’s pretty dangerous … they know they’re in the safe hands because they know what they’re doing out there”.*


-
*J.B.*


In March 2020, at one of the “Hunters’ Meetings” over coffee in the front foyer of the band office, S.A., J.B., and a handful of other experienced hunters immediately recognized the threat that the COVID-19 pandemic presented in terms food access for the community. J.B. recalls:
Basically, when all this planning started, me, Stan and some other hunters, we’re usually in the porch there, discussing everything like hunt-wise, and the planning and all that. So we were basically ready for whatever, yeah. So, all the planning just went from there… We knew eventually it [COVID-19] would make its way down here. We were already planning for what to do if it gets here, when it does get here. So … we’ll just try and get as much stuff [traditional food] as we can out there. There were times, like, it was almost every day hunting trips.
They developed a team of seven hunters, four of which were experienced hunters and three young learners. J.B. explained:
Yes, the younger guys, they wanted to volunteer. Yeah, so the older guys that usually go out hunting, offered their time to teach them and see what to prepare for, and what they’re doing is very important, especially with the pandemic.
The team went out on the land that spring of 2020 and, with compensation support from the band council, continued their hunt, nearly daily, for the better part of the year.

Organizing a safe and effective hunt requires experienced coordination to schedule the flights which bring hunters, their equipment, and food supplies in and out of their hunt camps. This usually requires 3–4 days of all day flights back and forth. During the pandemic, this was made more complicated as the flight coordinator had to ensure that the charter planes hired required their pilots to meet the same COVID-19 protocols that were set by Wapekeka.

The traditional food harvested varied with the seasons. Spring brought geese and caribou. Summer brought fish such as sturgeon and small game including partridge and beaver. The fall and winter brought moose, beaver, and caribou. Increasing the number of hunters along with the frequency and duration of the hunting trips resulted in a much greater yield than had been seen in years prior—at least for some of the main species. Geese harvesting was certainly an exception as the hunters were only able to harvest a similar amount that family harvests typically bring back in the spring. J.B. explained, “It’s changing, especially with the global warming, everything’s changing. Our numbers are just getting fewer and fewer every year—the migration is later.” In contrast, the fall and winter moose hunts were especially productive during the pandemic:
Our harvest was significant. I think that fall we had this many moose [made a wide gesture with his hands to show that it was a lot], plenty of it to go around…one of our better years I think.
Compared to the yield in other seasons, J.B. said with confidence “Oh, winter is way better.”

One reason the moose hunt is so successful in the winter is that hunters can access more territory by snowmobile over frozen waterways and land. Early in the season, right after freeze up, the bulls still have their antlers and stick to wide open areas which makes them easier to access. Later in the season, the moose drop their antlers and are much more proficient at moving through thicker brush. This adds a challenge and danger for the hunters, requiring them to follow the moose’s tracks through the dense bush of the forest. J.B. explained:
There’s no trails where the moose goes! They can go pretty fast, but you have to be careful because there are times when we flip our machines out there and smack into trees. So, you gotta be cautious when you’re out there; it can be dangerous.
Although the winter hunt usually brings in greater amounts of moose meat, it also requires lots of experience and skill to do safely. J.B. explained:
So you always gotta be prepared for everything, especially in 2019 or 2018, we were just like, kind of in a shock there. We were still living through the shock we had. We were [pause] what can I say, we were pretty wary, because of that hunting accident that happened. So that was still living with us in our minds, and we did manage it, so we kept on talking about it, be safe, be safe. We would have as many as 7, 6 people, so we usually say just make sure you bring everything extra. Extra boots, extra winter boots, rubbers, for when it gets slushy, like rubber boots, so that’s what they were dealing with too just slush on the lakes—it was just [pause] snow machines getting stuck one after another.

It takes many years and a lot of guidance to learn how to hunt effectively, efficiently, and safely. Slush and unstable ice conditions can unexpectedly add to the duration of a hunting expedition. Getting stuck in slush is related to climactic changes as well as the heavy weight of moose and other supplies that hunters need with them. While moose are often field dressed, during the pandemic, the hunters were trying to harvest as many moose as they could in the limited daylight. Amongst the six to seven hunters, three to four moose can be carried back. J.B. explained:
Yeah, usually when I would go hunting with them, I would carry a whole moose on my sled. Because we are still searching after we get a moose early, so we take advantage of the daylight. So we’re still searching. I’m usually in the back when I carry the load because the trail’s already packed.
This extra weight, combined with the changing snow and ice conditions, can increase the likelihood of getting stuck in slush. J.B. shared a story of one particularly long and arduous night where the team, with a few large moose in tow, were getting stuck in the slush repeatedly. The team was traveling in single file, as is common practice, so the hunters could stick together, keep track of each other, and help each other out if they got stuck. Jonas said that as a rule, “we never leave anybody behind.” Yet, on this night, they had to make the difficult decision to either keep a new hunter out while they worked on a stuck skidoo, or to send him home on his own. His extra clothes had gotten wet, and his body temperature was dropping rapidly. They had to weigh the risk of him traveling alone against the risk of him becoming hypothermic. Since removing a skidoo from slush is a lot of work with no guaranteed timelines, they decided as a group that keeping the cold young man in the bush brought greater risk. The hunting team decided to send him home on a lightened skidoo on his own. Making these kinds of decisions requires immense field experience and intimate knowledge of the hunters on your team. In this case, the older more experienced hunters stayed out late into the night to dig the skidoo out. They communicated with the community via their inReach©, a satellite texting device. Although the community was worried, they were reassured by the fact that the hunters were building fires and doing what they could to warm up. In the field, J.B. recalls that the hunters were calm. He explained, “they know they’re in safe hands because they know what they’re doing out there.” Ultimately, they were successful, and everyone came back safely. Yet, experienced hunters understand that this is not always a guaranteed outcome.

A tremendous amount of cultural knowledge is required to safely travel through the boreal forest, especially in the winter. Wapekeka’s careful maintenance and growing resurgence of this knowledge is significant considering the colonial legislation, practices, and institutions, such as residential schools, that worked to separate Indigenous youth from their families, repress cultural practices, and disconnect communities from free movement through and governance over their traditional lands. These massive disruptions to the dissemination of land-based knowledge in northern Ontario have been pointed to as a challenge for communities attempting to strengthen their local food systems, food security and food sovereignty [55,62,63,64]. In this context, Wapekeka’s harvesting efforts are significant as not only did the hunters help meet the community’s food needs during the pandemic, but the energy that experienced hunters extended to train new younger hunters builds community capacity to work towards meeting long-term access to traditional food.

#### 3.2.2. Food Preparation and Distribution


*“What should it look like, as a woman?”*


-
*Sophie McKay*


The hunters were not alone in their food provision concerns. This was on Sophie McKay’s mind too. Sophie McKay, a member of the Wapekeka Pandemic Committee, was responsible for community updates. As the COVID-19 pandemic hit and continued, she started to think about how a lengthy pandemic and limiting travel in and out of the community would impact access to food. She approached Priscilla Anderson, a community member who, although not involved in the Band Council, is a leader and champion of community initiatives. Priscilla says she is “the kind of person that helps people”. Sophie reached out to ask for her help in coordinating the processing, preparation, and distribution of food support to the community. Priscilla recalled her impressions of the early days of the pandemic.

I’m not sure when I first heard about it, but I didn’t think anything of it at first, but when I started hearing stuff about people getting sick and-you know-when they passed on, that’s when I started thinking about it, like thinking about the community, especially the children and the Elders. Those were the ones that I was worried about.

Sophie recalled asking Priscilla: “what should it look like as a woman?” While in this context, it is typical for men to hunt and women to process the food, there are many examples of women in the community who are active hunters. Sophie’s words “what should it look like as a woman”, importantly highlight the critical role women play, not only in preparing food, but in taking care of community needs at an even broader level.

Both women decided that they would work with the team of hunters to process and store the food they harvested in freezers. However, they also knew that traditional food alone would not be sufficient in meeting the needs of the community. They decided to focus their efforts on supporting both traditional food processing and distribution as well as the ordering, preparation, and distribution of market food. Knowing that each household had different needs and means, they wanted all food to be free to everyone. They decided to order enough for every household in the community to receive bi-weekly hampers of market and traditional food as well as other household sanitation and cleaning supplies.

The groceries were flown in on freight planes from Winnipeg. When the planes would land, Priscilla explained:
There were volunteers that went to meet the plane, rangers, and other volunteers, and they took the groceries, the bulk meat to the gym. And that’s where we were, just getting ready to help out and to organize—where are we going to put the stuff, like the groceries, the meat. It was a lot of work. And that was like, two buildings…It was a whole day thing…sometimes we were there like 14 h.
Once the gym and memorial center were set up, a member from each household was invited to pick up items at specific times to ensure numbers did not exceed community limits. For those community members who were unable to physically collect their groceries, the women created and delivered hampers to them. The hampers contained a collection of different foods and household items—meats, fruits, vegetables, household cleaning, sanitation products, snacks for the kids, and a Ziploc bag of traditional food from the hunters. The items were distributed based on household demographics. Although they were able to round up about ten volunteers, due to COVID-19 protocols, only five were allowed in the gym at one time which increased the duration and workload of their shifts.

Providing a combination of market food and traditional food was a real priority for community leadership and those involved in harvesting, preparing and/or distributing the food. Traditional food preparation and distribution required considerably more time and physical labour, as well as extensive coordination between everyone involved. Priscilla describes the time this work would take:
Sometimes the hunters would be out all day hunting, moose hunting, and one of the hunters would contact us to tell us what time they’re arriving and we would just go to the gym and wait for them and get the stuff ready, like, where to put the meat, and sometimes we just stayed there and butchered the meat. And then the next day we would continue. Oh my gosh, sometimes I would just go home, sleep, wake up, then we would go, keep doing the same thing every day.
The level of preparation and activity depended on the season and the type of wild game being harvested. Priscilla told us: “During the spring goose hunt, we prepared at least over 100 geese. It was a *lot* of work, not all at once but we had geese from—I’m not even sure—at least three camps.” When the hunters brought moose meat into the community in the fall, the women would begin processing immediately, regardless of what time the hunters came in. This would sometimes require them to work well into the night. The moose that were harvested in the winter would come in frozen, giving the women time to wait and begin processing the following day. Every fish, goose, or moose butchered would be led by older and more experienced women teaching a couple of younger women. The more experienced women would teach the younger women what to do—from how to hold the knife to how to store and distribute the food. Overall, the combined efforts of the hunters and the processors provided each household with a Ziploc bag of traditional meat every two weeks. They were also able to share some of their meat with community members in KI. On some occasions, volunteer drivers would meet at the halfway point of the roadblock between the two communities, exchanging Wapekeka moose meat for KI fish.

The women were also in charge of coordinating the distribution of market and traditional food. This required an intimate knowledge of each household and their individual family needs. Sophie told us they developed a chart which had information about each family and led the distribution of items as follows:
Everybody would get the same items, except some, like the ones that have toddlers, babies, we would add formula, pampers [diapers], wipes, baby shampoo, whatever the baby needs. That’s what we included in some of the hampers. And for the large families, we would add more because they had a big family.
People who ran out of food in between hamper distribution weeks could get refills, but Sophie’s master chart helped to make sure that everyone got something first before someone else could get more. Sometimes, people would contact Priscilla in the night asking for a refill, as they had run out of food and baby supplies. She would then go to the gym and put a hamper together for them, stating that, “if they’re in need, if they’re out of stuff…we just helped out”.

The food efforts described here continued for a year, from spring 2020 to spring 2021. To keep this effort up, Priscilla and Sophie were working 12–14 h days, seven days a week. This took a physical toll on Priscilla’s body. She explained:
Doing all that work, it made my back pretty sore. Sometimes I would go to the nursing station to get a shot to just ease that pain, the tension. I’m always—me not being 100%—I still showed up every day to help… I think it’s when the loads of freight came with the groceries, doing the lifting and stocking them where they needed to be. Yeah. Putting the meat in the freezer; it’s the lifting part that was hard for me. But I still did it.
Sophie and Priscilla expressed so much gratitude for the women who volunteered at different points throughout the year. Both women recognized that to take care of the community, the women volunteers had to leave their own children at home, and many of their husbands had to take over the work of childcare. Often, this would mean that women would leave the gym after a long shift of food organization and distribution to go home to manage their own households. This demonstrates the integral roles that women play in food procurement, a role that is often overlooked in research [54,56], even though women tend to bear a larger burden of food insecurity in households [65].

When Priscilla looks back, she laughed and said, “I was just so impressed with the hard work we did… Just looking back at the stuff we did, barely getting any sleep, we were just running on fumes.” Priscilla was not the only one who was impressed. People from the community expressed their gratitude as well, which was evident in one Facebook post:

My wife and I are so grateful for your endless work. We feel lucky to have people like you looking after us during this pandemic from hygiene supplies to providing food for us. Food given to us we respect, we cook or heat up, do don’t let it spoil, and store away for later use or eat. From bottom of our hearts. We thank and you and we feel lucky we have people like you.

Food mobilization efforts slowly began to wind down by summer 2021 with the gradual uptake of vaccinations in Wapekeka and in nearby communities. One Wapekeka Pandemic Committee member was impressed that despite some initial hesitancy, Wapekeka’s vaccination rate is currently at around 90% for eligible adults. They credit the individuals in the community for this success, citing that protecting Elders and children were the main motivators for people to take the vaccine. They were surprised to see some of the most ardent opposers to the vaccine ultimately decide to get it out of a desire to protect the most vulnerable in the community. This is another example of a contrast between the success of Wapekeka’s public health communication and the failures of much of Canada [66]. Strong leadership and a community desire to protect each other was a major factor in the success of Wapekeka’s pandemic response that was repeated in all conversations with community members.

## 4. Conclusions


*“I guess that’s just how Wapekeka is, eh?”*


-
*J.B.*


Wapekeka’s response to COVID-19 is a testament to the leadership, collaboration, and knowledge held by the community. The strength of Wapekeka’s community leadership was repeated in many conversations we had over coffee, dinner, and in our semi-structured interviews. A Wapekeka Pandemic Committee member confidently stated that “the leadership has been proactive and that’s what we need for every other community.” In each interview, participants were asked what it was about Wapekeka that enabled such successful collaboration to respond to a virus that was predicted to be devastating for remote and rural Indigenous communities. In each situation, people had difficulty articulating a specific reason. Perhaps J.B. summed it up best when he said with a smile, “I guess that’s just how Wapekeka is, eh?” In reality, it is most likely much deeper than a list of attributes or characteristics of a community and is perhaps so second-nature to the members of Wapekeka that it can feel like nothing special at all. It can feel like an obvious way to lead a community through a crisis. Indeed, the COVID-19 pandemic was an opportunity for many communities to reflect on the strength of their relationships, how they come together or not, and how they protected the most vulnerable or not.

While this story is a formidable example of individual and community effort to support and take care of themselves, it also highlights some of the ongoing challenges of land-based food procurement in the region. One challenge is the incredible cost of traditional food harvesting. A hunter must invest thousands of dollars in equipment, including float planes, skidoos, boats, seasonal clothing, nets, guns, ammunitions, etc., in order to navigate the land safely all year. It is a huge economic expense that without external supports, is not viable for many people in the community. Over the years, the hunters have needed to travel further and further from their hunting camps to access food sources, which increases costs dramatically. Policies and programs which aim to support communities in these efforts must truly understand what is involved in all aspects of traditional harvesting if support mechanisms are to have any meaningful impact.

This adds an important challenge to increasing traditional food in local diets, especially when taking the compounding challenges of environmental changes into consideration. While there is literature that demonstrates that Indigenous communities have developed complex wildlife management strategies over their extensive relationships with ecosystems in ancestral lands [67,68,69], current research contends that unprecedented levels of environment change in northern Canada present significant challenges to the resiliency of land-based food practices and the ecosystems they rely upon [65,70,71,72,73].

All the hunters we spoke with in informal conversations and in interview contexts raised concerns about ecosystem pressures related to harvesting rates and climate change. The uncertainty associated with the pandemic added further pressure to bring in as much meat as possible to sustain the community, yet hunters were concerned about moose population sustainability at such a harvesting rate. Training more hunters, processors, and distributors could increase yield, yet the hunters recognize the ecosystem’s limits and the importance of developing and employing their own protocols and systems of sustainable harvesting. There are also limits in terms of human labour. The physical and emotional labour involved in hunting, processing, and distributing food was immense, and tended to fall on the shoulders of a handful of people from the community.

Discussing the “enlightening” aspects or “silver linings” of the COVID-19 pandemic risks not fully acknowledging the devastating, divisive, and enduring impacts it has rendered globally. Yet, for a community that has been working towards strengthening local food harvesting, processing, and distribution for years, the COVID-19 pandemic was an opportunity to demonstrate exactly how much food could be brought in within the limits of the boreal forest ecosystem, economic resources, physical bodies, and individual motivation. The response to this pandemic demonstrated to the community their own capacity for harvesting traditional food. The pandemic also gave an opportunity for the experienced hunters and food processors to share their skills with the younger generation. Increasing youth interest in learning these skills has been a community goal for many years. J.B. noticed that during the pandemic, the community got to see and experience the value of being able to share traditional food, and as a result:

There’s a lot of interest now, especially these new kids talking about it. This is what we’re doing. Yeah, there’s more and more kids wanting to go camping too. So that’s why we had a pretty good number of campers this year, all kids.

There are many barriers for young people to get on the land and the opportunities provided through economic support did remove some of these barriers. This not only points to the importance of training, but also providing the economic means for the youth to get on the land in order to access that training. Observing what the lead hunters, processors, and community leaders were capable of, reified the continuing relevance of these traditional harvesting skills in a real crisis. It valorizes this way of life that has not been part of Ontario’s education systems and exposes the importance of acquiring skills that are critical for life in the north. Furthermore, seeing how important these efforts were for community health and wellbeing might be further motivation for younger people to learn these skills in order to take care of their own families and community in the future.

Overwhelmingly, chatting with community members and leaders in the community about their response to COVID-19 was accompanied by an immense sense of pride. Under the uncertain and stress-filled situation of the pandemic, the community witnessed: (1) proactive leadership that collectively established safe and effective COVID-19 protocols that limited COVID-19 spread and serious illness; (2) tremendous efforts of hunters who went on the land almost every day, in often brutal conditions, to bring back food to the community; and (3) herculean efforts of the women who organized market and traditional food preparation and distribution for the entire community. It is an example of what is possible when Indigenous communities—that are too often pathologized by the media, outsider researchers, politicians, health care professionals and educators—have greater autonomy over local governance, which is built on local values and strengths. This strength-based response from the Wapekeka First Nation highlights the ingenuity and perseverance of a people not unfamiliar with trauma but who continue thrive on “what we have”, and “what we can do with what we have.” This does not suggest that the community does not face critical vulnerabilities: there are housing shortages and overcrowding; there are limited and grossly underfunded health care services; there are critical food security and economic challenges, all of which contribute to ongoing health disparities. Despite this, Wapekeka First Nation’s community-led response to the COVID-19 pandemic highlights the ways that a community’s own knowledge and leadership are imperative in response to health crises. It requires the leadership in the community to navigate the often murky waters of integration between traditional food systems and health practices with western market food systems and medical support.

Despite real and perceived vulnerabilities to COVID-19, Wapekeka First Nation’s community-led response effectively protected the community. The emphasis on community collaboration and collective responsibilities guided the Wapekeka Pandemic Committee’s public health messaging and initiatives designed to support families through difficult periods of lockdown. While an outbreak of COVID-19 was a massive risk to the community, so were the measures to keep the virus out. Although lockdowns did limit individual mobility and made food security more precarious, the overall compliance of the community to the directives from leadership is astounding when compared to other community responses around the world. While there are probably many factors for this, one that stands out is the community’s emphasis on collaboration, reliance on cultural and traditional knowledge, and the centering of care for each other. While new waves and strains of COVID-19 continue to emerge across the globe, Wapekeka continues to keep cases out [74]. This story is another example of what IM and CBPR scholars and activists have been saying for years: communities already know their own challenges and strengths and are the most qualified to determine their own solutions. The decision to coordinate food hampers, purchase freezers, and support hunters to harvest all year long was a community decision, not an initiative driven from the outside. Wapekeka’s story proves an important model of Indigenous “survivance” and sovereignty [8], and an example of how to keep a community together through a crisis that in so many places, drove people apart.

## Figures and Tables

**Figure 1 ijerph-19-11562-f001:**
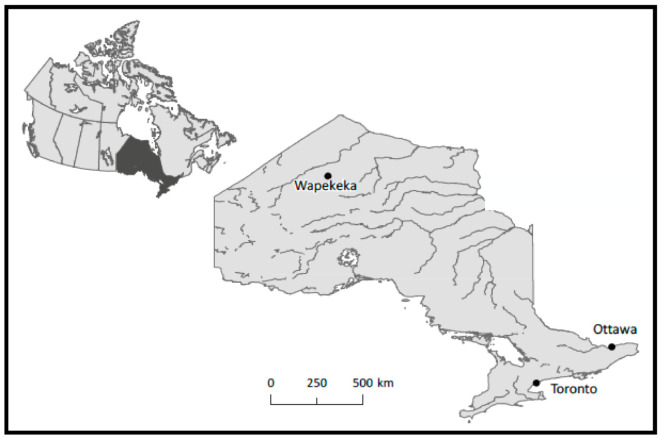
Map identifying the location of the First Nation community of Wapekeka in northern Ontario, Canada. Map created by Sarah Simpkin, University of Ottawa Library, on 1 November 2018 using Natural Earth data in ArcGIS Desktop.

## Data Availability

The data is contained within the article.

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
