# Peer review of "Wapekeka’s COVID-19 Response: A Local Response to a Global Pandemic"

_ijerph, 2022, doi:10.3390/ijerph191811562_

Round 1
Reviewer 1 Report
Please describe the methodology section of the study and include the discussion section
Author Response
Please describe the methodology section of the study and include the discussion section
Thanks so much for taking the time to review our paper. Some other reviewers have given particular suggestions for updating our methodology to be more descriptive, such as including exactly how we used Facebook to connect with community members and to be specific about why we used particular methodologies. We feel with these suggestions our methodology section has been improved. As per the discussion, it is actually included in the results. We felt that organizing the sections this way improved the readability and suited the story-telling of the interviewees better. It does deviate slightly from traditional academic papers, but we feel that this deviation serves the community’s story better than separating it would.
Reviewer 2 Report
This is an excellent article and work overall because of the Indigenous community-based approach, collaborations, and foundations. The voices of the authors and Wapekeka community and COVID-19 responders, including hunters, food processors, volunteers, and food sovereignty leaders among others, are the greatest contribution of this project and publication. The heaviest concern and recommendation for this article to grow is how the quotations from the wonderful interviews are introduced, contextualized, and cited. There are various instances where the words of the authors and the interviewees blur, and it can be confusing to the readers. Who is speaking? The readers face this question about the source of the voice and reference in different parts of the articles. There are also a few minor distracting typos that can be addressed for the writing clarity, including the citation format and style.
Some more specific questions and feedback include the following:
On page 4, the authors refer to social media and various forms of communication such as Facebook and text messages. Is there a way to cite or include some more examples from these communications that informed the project or explain more about how they communicated in these ways? For example, it is interesting how important Facebook is. Could the significance of Facebook and how it is used be explained more? How many people have Facebook from the community? How accessible is it?
Even if the citations come from anonymous interviewees, I recommend that citations be included and simply note when the interviewee is anonymous or has a pseudonym. Some quotes are inserted in the text without an introduction (such as the beginning of page 6), and it is hard to follow or know the source of the quote (or even to recognize it is a quote).
When possible and appropriate, could you introduce the interviewees more? What is their background to contextualize their words?
On page 6, there is a reference to how the Canadian government sent body bags to First Nations in response to their plea for help during the H1N1 pandemic, which reminds me of how a similar case happened in the U.S. during the COVID-19 pandemic that might be worth mentioning. See the case of the Seattle Indian Health Board in 2020: https://www.nbcnews.com/news/us-news/native-american-health-center-asked-covid-19-supplies-they-got-n1200246.
On page 9, there is a reference to a hunting accident. Could you contextualize or share some background about the hunting accident if possible or explain more?
Climate change is a major theme throughout the manuscript. It could help to provide more emphasis and context about climate change and its impacts in the specific context of the Wapekeka First Nation and their homelands.
Gender is also an important aspect of this article. It is unclear if hunters can only be men and what these different gender roles mean to the Wapekeka Indigenous community and people. Considering page 11, for instance, were the gender roles in food sovereignty, hunting, and preparation based on Indigenous traditions? Or were there major changes over time, especially in the context of the COVID-19 pandemic? On page 13, could you elaborate on what you mean regarding a “gendered lens” to understand climate change?
Education is key in this article too, such as how the experienced elders teach the younger generations, which could be framed more in the manuscript.
Thank you for this powerful work that shows the strength and brilliance of the Wapekeka and diverse Indigenous people, which inspires and sustains hope in humanity even in the most difficult challenges and time.
Author Response
Thanks so much for taking the time to review our article and for your important feedback and suggestions. We have addressed specific and general concerns below.
The heaviest concern and recommendation for this article to grow is how the quotations from the wonderful interviews are introduced, contextualized, and cited. There are various instances where the words of the authors and the interviewees blur, and it can be confusing to the readers. Who is speaking?
Thank you so much for pointing this out. Upon review of the manuscript given to reviewers, we see that it is in fact a formatting error – in our original manuscript we used block quotes to indicate the voice of the interviewee. We completely understand how confusing that must have been to read. We have corrected by reformatting with block quotes and we hope that this helps to clarify this confusion.
There are also a few minor distracting typos that can be addressed for the writing clarity, including the citation format and style.
Thank you for this. We have reviewed the manuscript and corrected these errors, including the citation format and style.
On page 4, the authors refer to social media and various forms of communication such as Facebook and text messages. Is there a way to cite or include some more examples from these communications that informed the project or explain more about how they communicated in these ways? For example, it is interesting how important Facebook is. Could the significance of Facebook and how it is used be explained more? How many people have Facebook from the community? How accessible is it?
These are really interesting points. We have included more specifically exactly how Facebook was used in this particular response to COVID (which Facebook pages and groups, etc). We also included a sentence just to point out the popularity of Facebook as a communication tool in the north. The points you raise about accessibility are also important, however, we feel we would need to conduct further research alongside the community to comment on its accessibility to community members.
Even if the citations come from anonymous interviewees, I recommend that citations be included and simply note when the interviewee is anonymous or has a pseudonym. Some quotes are inserted in the text without an introduction (such as the beginning of page 6), and it is hard to follow or know the source of the quote (or even to recognize it is a quote).
We hope that addressing the formatting errors as indicated above improves this.
When possible and appropriate, could you introduce the interviewees more? What is their background to contextualize their words?
This is another important point. Based on community protocols, we introduced interviewees as much as they deemed appropriate by stating their relative positions in the community and roles in the pandemic response.
On page 6, there is a reference to how the Canadian government sent body bags to First Nations in response to their plea for help during the H1N1 pandemic, which reminds me of how a similar case happened in the U.S. during the COVID-19 pandemic that might be worth mentioning. See the case of the Seattle Indian Health Board in 2020: https://www.nbcnews.com/news/us-news/native-american-health-center-asked-covid-19-supplies-they-got-n1200246.
Thanks, this is a really important connection. We decided to keep our paper and critique focused on the Canadian federal and provincial governments for the scope of this paper but there are clear connections across the border.
On page 9, there is a reference to a hunting accident. Could you contextualize or share some background about the hunting accident if possible or explain more?
Due to the sensitivity of the accident, out of respect and privacy, we shared as much as we were able to. Our purpose for stating it was just to point out that food harvesting can be quite dangerous and accidents happen even to the most experienced hunters. It was to illustrate that safety is taken very seriously by the team and that there is great risk involved in harvesting from the land. We hope that this purpose comes across without having to go into too much detail.
Climate change is a major theme throughout the manuscript. It could help to provide more emphasis and context about climate change and its impacts in the specific context of the Wapekeka First Nation and their homelands.
This is another excellent point and it is very true that this is impacting the communities. However, for the scope of this paper, we felt that adding a few references would serve the paper in a better way than going into too many details. While climate change is definitely important, it acts more as another pressure in the background of the community’s response. The authors felt that adding more detail here could distract from the narrative.
It is unclear if hunters can only be men and what these different gender roles mean to the Wapekeka Indigenous community and people. Considering page 11, for instance, were the gender roles in food sovereignty, hunting, and preparation based on Indigenous traditions? Or were there major changes over time, especially in the context of the COVID-19 pandemic?
This is an excellent point. We have added a few sentences to address the traditional and contemporary gender roles, and the nuances in these delineations and changes. For example, we noted that in this community, men do typically hunt while women tend to process the harvested food. However, there are several examples of women who hunt - including the daughter of one of the main hunters in this story.
On page 13, could you elaborate on what you mean regarding a “gendered lens” to understand climate change?
Thanks for pointing this point out. We determined that this sentence really is incomplete and that as valid as this description is, it requires a much more thorough explanation that goes beyond the scope of this paper.
Education is key in this article too, such as how the experienced elders teach the younger generations, which could be framed more in the manuscript.
Thank you for pointing this out, we completely agree with you. We have tried to draw more attention to this in the sections it is brought up.
Reviewer 3 Report
Thank you for the opportunity to review this deeply respectful, insightful and strengths based article that discusses First Nations’ communities response to COVID 19. Providing a colonisation and historical context, identifying the limitations of Treaty agreements, and remote community experiences and Government responses of previous epidemics makes this a very strong paper. Being written alongside two hunters for the community who hold recognised positions within their communities, and providing longer quotes from community members ensures that this paper gives community members a voice in highly respectful and authentic ways.
Abstract
Well written, clear.
Introduction
Well written, sub headings help signpost the introduction
Can you explain what a boil water advisory is, for an international audience.
Research methods and methodology
Clearly outlines the process undertaken and why.
Indigenous methodologies (IM) and community-based participatory research (CBPR) references could also be inserted earlier in the paragraph.
Is it possible to provide a reference for OCAP? Particularly for an international audience who may wish to read about it further.
I was unsure the cultural background of all authors, particularly those with the first 2 affiliations.
Jonas is described as the forth author, but is listed as third. (p5, line 203)
Results and Discussions
It makes sense to present both results and discussion together in this paper. The story telling approach works well and reflects Indigenous methodologies.
The longer quotes from community members are relevant, but may be read more easily as such if they were italicised so that it is clearer which are direct quotes, and which are author’s words, unless the two were purposefully blended because the community members are also authors?
The writing style enables deeper insights and understanding, and is incredibly strengths based – these are very important aspects of this paper. Describing both men and women’s roles is very powerful.
Describing what happened coming out of the initial Pandemic and the uptake of immunisation completes the story, emphasising the importance of leadership.
Conclusion
This section finishes the paper strongly. I wondered if it were discussion r conclusion, but in reviewing what is discussed in the results and discussion section, can understand what the authors are doing in the conclusion section.
Referencing
I note that many of the references are older, but in the context of historical events, they are relevant.
However, I did consider that reference 34 Mowbray Social Determinants and Indigenous Health: The International Experience and Its Policy Implications could be updated to, for example
George, E. , Mackean, T. , Baum, F. , Fisher, M. (2019). Social Determinants of Indigenous Health and Indigenous Rights in Policy: A Scoping Review and Analysis of Problem Representation. The International Indigenous Policy Journal,10(2) . DOI:10.18584/iipj.2019.10.2.4 https://ojs.lib.uwo.ca/index.php/iipj/article/view/8059/6589.
One of these authors is an Indigenous woman.
Author Response
Can you explain what a boil water advisory is, for an international audience.
Thank you for pointing this out. We added this in an end note.
Clearly outlines the process undertaken and why.
We were more explicit about why we wanted to visit the community in order to get a better picture of what happened during COVID-19. We wanted to make sure that Stan and Jonas were authors as this story is not our story, it is the community’s story. We specifically asked the women what their role was as it is often missing in research on food security and sovereignty.
Indigenous methodologies (IM) and community-based participatory research (CBPR) references could also be inserted earlier in the paragraph.
We have adjusted where these are added.
Is it possible to provide a reference for OCAP? Particularly for an international audience who may wish to read about it further.
Yes, thank you for pointing this out.
I was unsure the cultural background of all authors, particularly those with the first 2 affiliations.
We have described the background of the second and third authors in the abstract and in the introduction. We added the cultural backgrounds of the other first, third, and fourth authors in an endnote . We chose to do it this way so as not to distract from the importance of the second and third author, and to emphasize that regardless of our backgrounds, we are all visitors
Jonas is described as the forth author, but is listed as third. (p5, line 203)
Thank you for pointing this out. We have adjusted this discrepancy in the manuscript.
The longer quotes from community members are relevant, but may be read more easily as such if they were italicised so that it is clearer which are direct quotes, and which are author’s words, unless the two were purposefully blended because the community members are also authors?
Yes, thank you so much for pointing this out, we are sure it must have been challenging to read. This was actually a formatting error that happened along the way – originally, we used block quotes to delineate when it was the interviewees' words. We formatted the manuscript again to include this. Thank you!
I did consider that reference 34 Mowbray Social Determinants and Indigenous Health: The International Experience and Its Policy Implications could be updated to, for example
George, E. , Mackean, T. , Baum, F. , Fisher, M. (2019). Social Determinants of Indigenous Health and Indigenous Rights in Policy: A Scoping Review and Analysis of Problem Representation. The International Indigenous Policy Journal,10(2) . DOI:10.18584/iipj.2019.10.2.4 https://ojs.lib.uwo.ca/index.php/iipj/article/view/8059/6589.
Thank you for pointing this out! We have changed the reference.
Round 2
Reviewer 1 Report
The manuscript has improved and is now correct for publication.